# Nickel-catalysed selective migratory hydrothiolation of alkenes and alkynes with thiols

Yulong Zhang[1,3], Xianfeng Xu[1,3] & Shaolin Zhu [1,2]

Direct (utilize easily available and abundant precursors) and selective (both chemo- and regio-) aliphatic C–H functionalization is an attractive mean with which to streamline chemical synthesis. With many possible sites of reaction, traditional methods often need an adjacent polar directing group nearby to achieve high regio- and chemoselectivity and are often restricted to a single site of functionalization. Here we report a remote aliphatic C–H thiolation process with predictable and switchable regioselectivity through NiH-catalysed migratory hydrothiolation of two feedstock chemicals (alkenes/alkynes and thiols). This mild reaction avoids the preparation of electrophilic thiolation reagents and is highly selective to thiols over other nucleophilic groups, such as alcohols, acids, amines, and amides. Mechanistic studies show that the reaction occurs through the formation of an RS-Bpin intermediate, and THF as the solvent plays an important role in the regeneration of NiH species.

[1] State Key Laboratory of Coordination Chemistry, Jiangsu Key Laboratory of Advanced Organic Materials, School of Chemistry and Chemical Engineering, Nanjing University, 210093 Nanjing, China. [2] State Key Laboratory of Bioorganic and Natural Products Chemistry, Shanghai Institute of Organic Chemistry, Chinese Academy of Sciences, 210032 Shanghai, China. [3]These authors contributed equally: Yulong Zhang, Xianfeng Xu. Correspondence and requests for materials should be addressed to S.Z. (email: shaolinzhu@nju.edu.cn)

**O**rganosulfur compounds, metabolites or macromolecules essential to life, are prevalent in pharmaceuticals, natural products, and materials (Fig. 1a)[1–3]. They compose ~20% of all Food and Drug Administration-approved drugs[4,5]. The development of protocols for the sustainable and efficient construction of C–S bonds is important in chemical synthesis. Commonly used methods for the construction of such bonds include Michael addition, $S_N2$-type alkylation, and the powerful transition-metal-catalyzed C–S cross-coupling[6–11]. One potential and more attractive strategy for their construction is through the selective C−H functionalization[9], because this leads to the utilization of more widely available starting materials or more concise synthetic routes. However, to achieve excellent regio- and chemoselectivity, most of these processes need a polar directing group in the vicinity, and this limits their application in organic synthesis. As an alternative, the recently emerging metal-hydride-catalyzed[12–15] olefin remote functionalization[16–53] can install a functional group at a distal position in a hydrocarbon chain under mild conditions. Starting from the ubiquitously available olefin-containing substrates, and using an extra hydride source, the NiH-catalyzed remote hydrofunctionalization[39–53] with aryl/alkyl halides as electrophiles has been established as a powerful protocol for the construction of a diverse range of C–C

bonds at a distal, inert $sp^3$ C–H position (Fig. 1b)[43–53]. However, the electrophilic amination or thiolation reagents required to forge the more challenging carbon–heteroatom bond are generally not stable and often not commercially available, especially when bearing functional groups. Their preparation is nontrivial and time consuming, and often involves the use of the stoichiometric amounts of hazardous reagents.

To address these challenges, we enquired whether unmodified nucleophilic thiols that are widely available could be employed directly. Here we present the successful application of these ideas and describe an operationally trivial approach that allows the direct selective $sp^3$ C–H thiolation with a naked thiol (Fig. 1c) at a distal benzylic position, the α-carbon of an ether, or a terminal position of the hydrocarbon chain of an alkene. A number of features of a transformation of this sort can be highlighted as follows: (a) high chemoselectivity of the thiol group in the presence of a series of potentially reactive functional groups such as amides, acids, alcohols, and amines; (b) excellent regioselectivity amongst multiple sites, including a benzylic position, a carbon α to the oxygen atom position, or a terminal position; (c) a regioconvergent process for the conversion, for example, of isomeric mixtures of olefins; and (d) feedstock thiols as thiolation reagents, such a process avoids the preparation of electrophilic thiolation reagents.

## Results

**Regiodivergent thiolation reaction design and optimization.** We began our investigation by examining the remote hydrothiolation of 4-phenyl-1-butene (**1a**) with benzyl mercaptan (**2a**). After careful evaluation of a variety of nickel sources, ligands, bases, hydride sources, and solvents (Fig. 2), we found that a reaction at 60 °C employing a combination of NiI₂ as a catalyst, bathocuproine (**L1**, 2,9-dimethyl-4,7-diphenyl-1,10-phenanthroline) as a ligand, HBpin (pinacolborane) as the hydride source, Li₃PO₄ as the base, and mixed tetrahydrofuran/acetonitrile (THF/CH₃CN) as the solvent delivers the desired migratory benzylic thiolation product (**3a**) in 75% isolated yield as a single regioisomer [regioisomeric ratio (major product: all other isomers) >99:1] (Fig. 2, entry 1). Use of other nickel sources, such as NiCl₂ or NiBr₂, leads to diminished

### a   Representative organosulfur compounds in natural products and pharmaceuticals

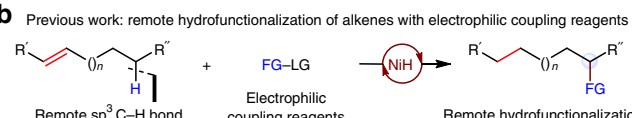

**Griseoviridin** (antibiotic nature product)

**Cinanserin** (schizophrenia/anti-SARS drug)

**Nelfinavir (Viracept)** (anti-HIV drug)

**Montelukast (Singulair)** (allergy and asthma drug)

**Cilastatin (Primaxin)** (antibacterial drug)

### b   Previous work: remote hydrofunctionalization of alkenes with electrophilic coupling reagents

Remote sp³ C–H bond   +   Electrophilic coupling reagents   →   Remote hydrofunctionalization

### c   This work: selective migratory thiolation of unactivated alkenes/alkynes with thiols directly

Direct thiolation of remote sp³ C–H bond   +   (Feedstock Nu-H) selective coupling of thiol   →   Chemo- and regioselective remote hydrothiolation

**Transformation highlights**

(1) Chemoselective: thiol (-SH) vs. amide(-NHCOR), alcohol (-OH), acid (-CO₂H), amine (-NH)

(2) Regioselective (multiple site-selectivities): benzylic, α-carbon of an ether, terminal, etc.

(3) Regioconvergent process: value-added conversion of isomeric mixtures of olefins

(4) Nucleophilic thiols as thiolation reagents: no preparation of electrophilic thiolation reagents

**Fig. 1** Design of a NiH-catalyzed remote hydrothiolation reaction. **a** Representative organosulfur compounds in natural products and pharmaceuticals. **b** Previous work: remote hydrofunctionalization of alkenes with electrophilic coupling reagents. **c** This work: selective migratory hydrothiolation of unactivated alkenes/alkynes with thiols directly. *tBu, tert-butyl; Ph, phenyl; FG, functional group; LG, leaving group

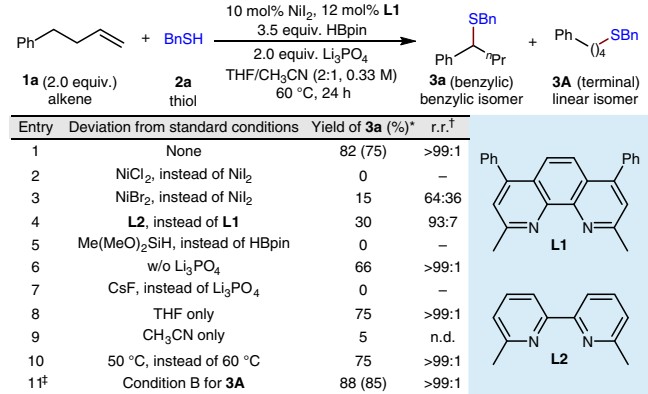

| Entry | Deviation from standard conditions | Yield of 3a (%)* | r.r.† |
|---|---|---|---|
| 1 | None | 82 (75) | >99:1 |
| 2 | NiCl₂, instead of NiI₂ | 0 | – |
| 3 | NiBr₂, instead of NiI₂ | 15 | 64:36 |
| 4 | **L2**, instead of **L1** | 30 | 93:7 |
| 5 | Me(MeO)₂SiH, instead of HBpin | 0 | – |
| 6 | w/o Li₃PO₄ | 66 | >99:1 |
| 7 | CsF, instead of Li₃PO₄ | 0 | – |
| 8 | THF only | 75 | >99:1 |
| 9 | CH₃CN only | 5 | n.d. |
| 10 | 50 °C, instead of 60 °C | 75 | >99:1 |
| 11‡ | Condition B for **3A** | 88 (85) | >99:1 |

**Fig. 2** Optimization of regiodivergent remote hydrothiolation. *Yields were determined by gas chromatography (GC) analysis using n-tetradecane as the internal standard. The yield within parentheses is the isolated yield and is an average of two runs (0.20 mmol scale). †r.r. refers to regioisomeric ratio, representing the ratio of the major product to the sum of all other isomers as determined by GC analysis. ‡The linear thioether (**3A**) is obtained as a single isomer; conditions B for terminal selectivity: NiI₂ (5 mol%), **L2** (6 mol%), (MeO)₂MeSiH (3.5 equiv.), DMSO (0.50 M), 50 °C, 48 h. Ph, phenyl; Bn, benzyl; nPr, n-propyl; HBpin, pinacolborane; THF, tetrahydrofuran; DMSO, dimethyl sulfoxide

yields and selectivities (Fig. 2, entries 2 and 3). Ortho substituents in the bipyridine ligand are critical for the reaction, and use of a similar ligand (**L2**) leads to inferior yield and regioisomeric ratio (Fig. 2, entry 4). Changing the hydride source to silanes, such as dimethoxy(methyl)silane, results in none of the desired product (Fig. 2, entry 5). The addition of the base Li₃PO₄ improves the yield but is not essential (Fig. 2, entry 1 vs. entry 6). CsF, which we previously used in remote hydroarylation reactions[45], leads to complete failure of the reaction (Fig. 2, entry 7). Notably, control experiments show that the cyclic ether solvent is necessary for the reaction to proceed (Fig. 2, entry 8 vs. entry 9). In addition, a slightly lower yield is obtained at lower temperature (Fig. 2, entry 10). Interestingly, after a thorough re-evaluation of the reaction parameters, we were able to change the thiolation site to the terminal position[54–60] to generate a very good yield of the linear thioether as a single isomer (Fig. 2, entry 11).

**Substrate scope**. With the optimal conditions in hand, we sought to define the scope of the alkene component (Fig. 3). First, an array of terminal aliphatic alkenes with a variety of *ortho*, *meta*,

and *para* substituents on the remote aryl ring (**3c–3l**) are found to perform well producing the desired benzylic thioether exclusively (Fig. 3a). Substrates containing both electron-rich (**3c** and **3g**) and electron-deficient (**3d–3f** and **3 h**) arenes are suitable for this reaction. Structurally complex aromatic systems such as sugar-linked aryl ring (**3j**) and camphor-linked aryl ring (**3k**) are amenable to the migratory cross-coupling. Heteroaromatic substrates, such as those containing a pyridine-linked aryl ring (**3l**) or a thiophene (**3m**) in place of the aryl group, are also well tolerated. Unactivated internal olefins also readily undergo alkene isomerization-hydrothiolation smoothly (Fig. 3b). As expected, *E/Z* alkene mixtures (**3n–3r** and **3t**) react well, and high selectivity for thiolation at the benzylic position is observed, regardless of the starting position of the C=C bond. For substrates with a tertiary carbon on a benzyl position, which previous reports[43,46,48] have noted as challenging, migration towards the benzylic position and subsequent thiolation to generate the S-containing tetrasubstituted carbon center is still preferred (**3t**). Styrenes themselves (**3u–3c′**) are also suitable partners under these conditions (Fig. 3c). Compounds with a variety of functional groups on the aryl ring of styrene are tolerated, including

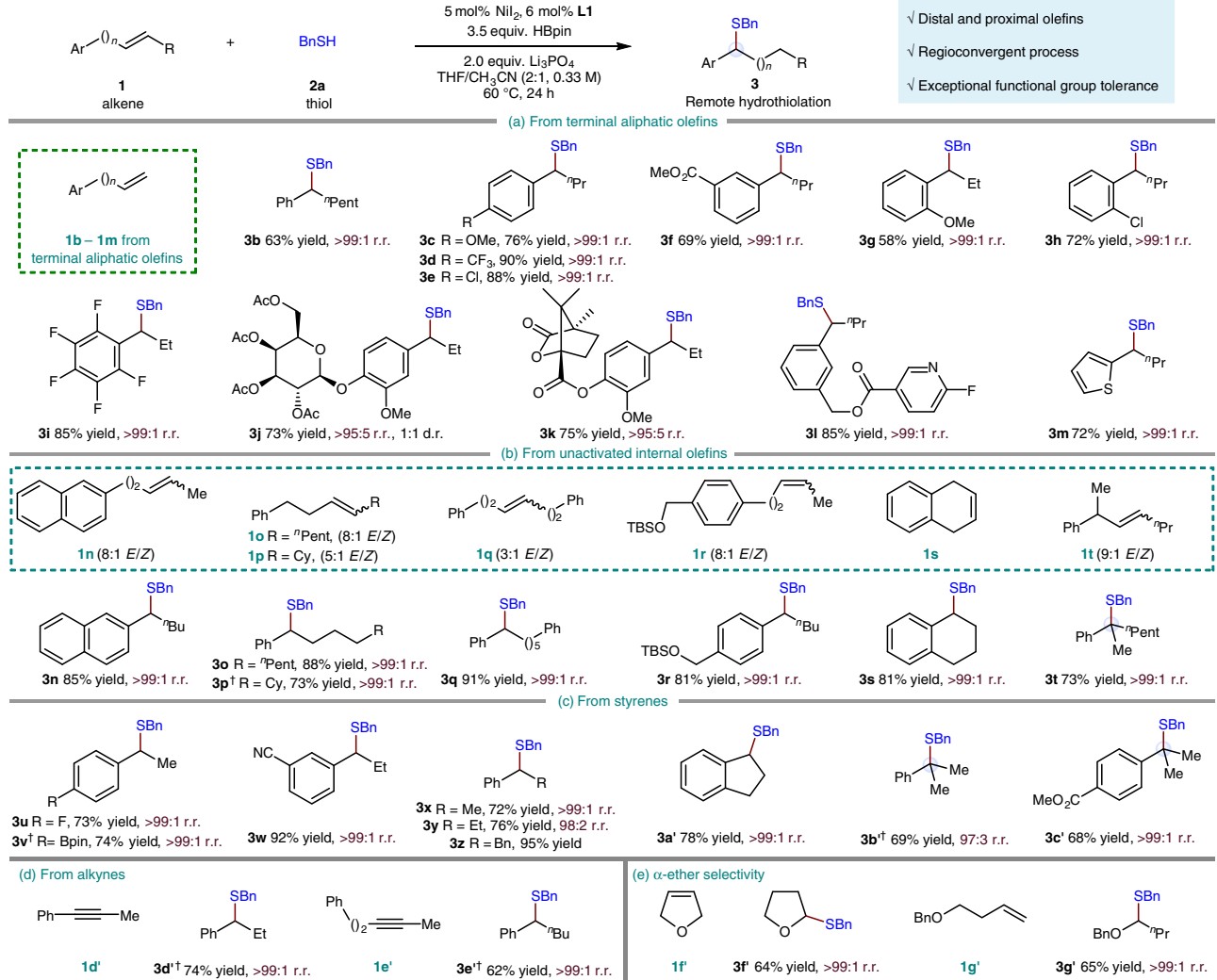

**Fig. 3** Substrate scope of alkene component. Under each product are given yield in percent, and regioisomeric ratio (r.r.). Yield refers to isolated yield of purified product (0.20 mmol scale, average of two experiments). r.r. represents the ratio of the major product to the sum of all other isomers as determined by gas chromatography (GC) analysis, ratios reported as >95:5 were determined by crude proton nuclear magnetic resonance (¹H NMR) analysis. †Forty eight hours. Me, methyl; Et, ethyl; ⁿBu, n-butyl; ⁿPent, n-pentyl; Cy, cyclohexyl; TBS, *tert*-butyldimethylsilyl

an aryl fluoride (**3u**), a boronic acid pinacol ester (**3v**), an aryl nitrile (**3w**), and an ester (**3c'**). The reaction can also be extended to α-methyl styrenes to provide exclusively the benzylic thioethers (**3b'** and **3c'**) with a fully substituted carbon center.

It is important to highlight that alkynes, another type of easily prepared starting material, could undergo reductive remote hydrothiolation to deliver the same migratory thiolation products (**3d'** and **3e'**, Fig. 3d). Mechanistically, the vinylnickel intermediate formed upon hydrometallation of the alkyne is selectively captured by a proton source (thiol) forming an alkene. Isotope labeling experiments indicated that the source of protons in this reaction is mainly from the thiol (see Supplementary Fig. 10 for details), while the alkylnickel intermediate formed upon hydrometallation of this alkene selectively engages with the NiH-catalyzed chainwalking-thiolation reaction. Finally, the current benzylic regioselectivity can also be easily extended, as in thiolation at the carbon α to the oxygen atom, producing the monothioacetals (**3f'** and **3g'**, Fig. 3e) in moderate yields as single isomers.

Further investigation of the reaction demonstrated the broad scope of thiol partner (Fig. 4). In general, both aliphatic (**4b–4m**) and aromatic (**4n–4c'**) thiols are excellent reaction partners and give the corresponding benzylic thioethers with good to excellent yields and regioselectivities. An array of primary and secondary aliphatic thiols all prove to be competent substrates, delivering the desired benzylic thiolation products in good to excellent yields (**4b–4l**). For the steric hindered tertiary thiol, a disulfide can be used to obtain a satisfactory yield (**4m**). In addition, a variety of electron-withdrawing (**4o–4u** and **4w**) and electron-rich (**4v** and **4x–4b'**) thiophenol derivatives are competent substrates. A variety of heterocycles frequently found in medicinally active agents, including both furan (**4i**, **4c'**) and thiophene (**4j**), are also compatible, and a variety of functional groups are readily

accommodated, including esters (**4e** and **4h**), an aryl fluoride (**4p**), an aryl chloride (**4q–4s**), and ethers (**4u–4x**). Notably, potential coupling motifs, including a primary alcohol (**4f**), a primary carboxylic acid (**4g**), a phenol (**4y**), a primary aniline (**4z**), a secondary Boc carbamate (**4h**), and a secondary acetyl amide (**4a'**) remain intact, which demonstrates both the excellent chemoselectivity of this transformation and their potential application in selective cysteine conjugation in biomolecules.

## Discussion

To gain some insights into the chainwalking process of olefin isomerization, olefin **1a** was subjected to the standard reaction conditions in the absence of any thiol. A significant amount of other olefin isomers arising from the olefin isomerization is observed within 1 h, which indicates that occurrence of olefin isomerization does not depend on the presence of the thiol and also suggests that olefin isomerization is unrelated to C–S coupling (Fig. 5a, above). Additionally, consistent with our previously reported results, a mixture of olefins is observed when the reaction is run to partial conversion (Fig. 5a, below), indicating that olefin isomerization proceeds with fast dissociation and reassociation of the NiH species. Furthermore, the corresponding isotopic labeling experiments were carried out with deuterothiol and deuteropinacolborane, respectively (Fig. 5b). No deuterium incorporation in the desired product is noted when deuterothiol is used, indicating that the thiol is not involved in chainwalking process. As expected, deuterium scrambling and deuterium incorporation is observed at all positions along the aliphatic chain, with the exception of the benzylic position. Mass spectrometric analysis revealed that a mixture of undeuterated, monodeuterated, and polydeuterated products is obtained. This is consistent with the hypothesis that chainwalking occurs with

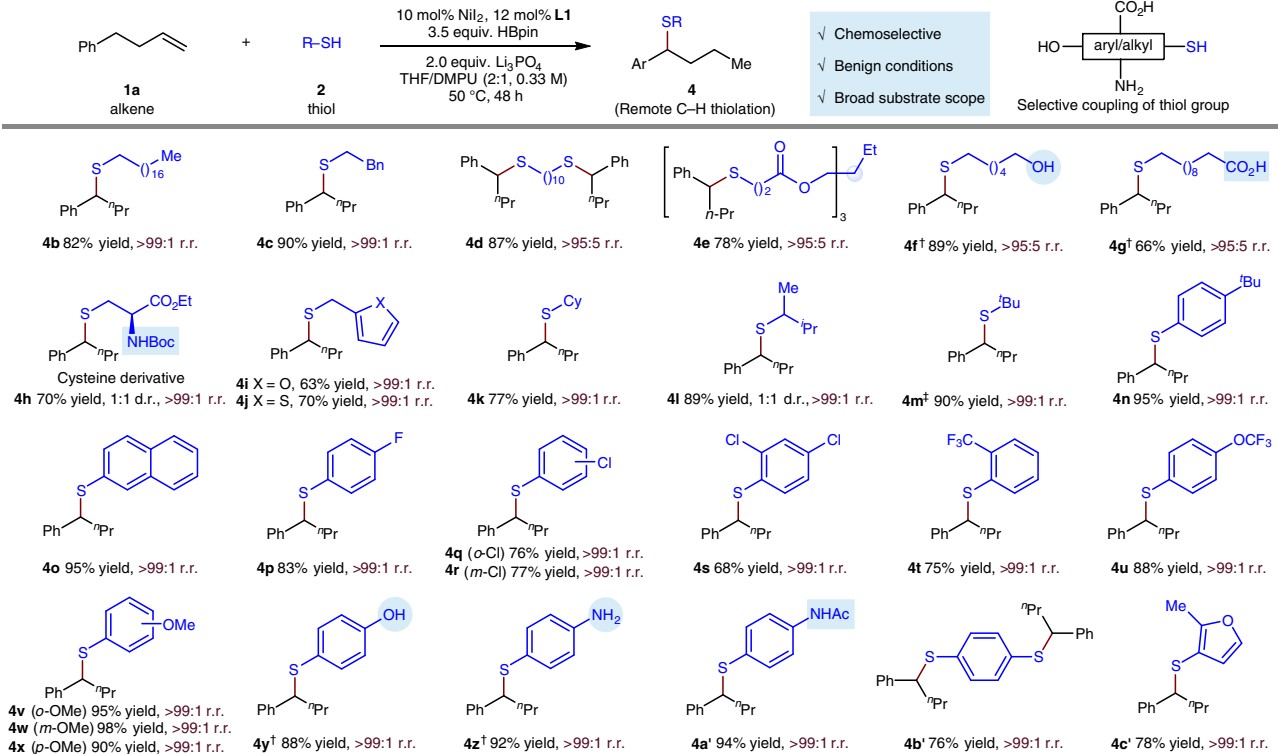

**Fig. 4** Substrate scope of thiol partner. Under each product is given yield in percent, and either the regioisomeric ratio (r.r.) or the diastereomeric ratio (d.r.). Yield and r.r. are as defined in Fig. 3 legend. †5.0 equiv. HBpin was used. ‡Di-*tert*-butyl disulfide (0.10 mmol, 0.50 equiv.) was used. Boc, *tert*-butoxycarbonyl; Ac, acetyl; DMPU, 1,3-dimethyl-3,4,5,6-tetrahydro-2(1*H*)-pyrimidinone

dissociation and reassociation of free NiH/NiD from the NiH/NiD-alkene complex. Finally, no migratory reaction takes place when the linear sulfide (**3A**) is resubjected to the standard

conditions, suggesting that chainwalking preceeds the C–S coupling (Fig. 5c, above). Following the detection of trace amounts of a remote hydroboration product (**6**), this migrated hydroboration intermediate was resubjected to the standard conditions. However, no desired thiolation product was observed, suggesting that the C–S coupling step does not proceed through the remote hydroboration intermediate (Fig. 5c, below).

To shed light on the thiolation process, a variety of experiments were carried out. When 0.5 equiv. of a symmetrical disulfide is used instead of 1.0 equiv. of the corresponding thiol, the desired remote hydrothiolation product is obtained in a comparable yield (Fig. 6a), indicating that the disulfide might be the potential reactive intermediate of the thiol. Monitoring the remote hydrothiolation reaction of a disulfide by $^{19}$F NMR (fluorine-19 nuclear magnetic resonance), however, indicates that the disulfide ($\delta = -114.9$ ppm) is first transformed into an RS-Bpin intermediate ($\delta = -116.8$ ppm) (Fig. 6b). Significantly, analogous experiments on the corresponding thiol substrate ($\delta = -118.9$ ppm) also reveal the generation of this RS-Bpin intermediate ($\delta = -116.8$ ppm) with no trace of disulfide detected (Fig. 6c). Meanwhile, the generation of $H_2$ in this standard reaction is also observed by gas chromatography (GC) analysis. Overall, these results reinforce the notion that the disulfide is not involved as the active intermediates from the thiol, and suggests that the in situ generated RS-Bpin might be the actual thiolation reagent.

Encouraged by these results, we wondered whether the pre-generated RS-Bpin reagent could be employed directly instead of a thiol. Indeed, as shown in Fig. 7, changing the thiolation reagent from a thiol to RS-Bpin **2A'**, generated in situ from the thiol and HBpin, a competent yield of **3a** is obtained. In this case, only a stoichiometric amount of the alkene is required and the desired thiolation product could still be obtained in comparable yield (cf. Fig. 7, entries 3–6). For instance, **3a** is obtained in 84% yield when

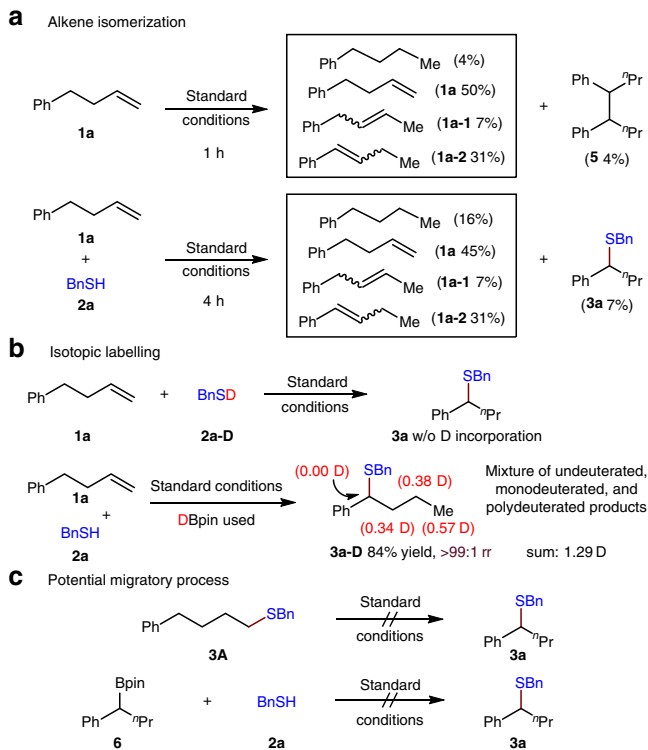

**Fig. 5** Mechanistic studies of chainwalking. **a** Alkene isomerization. **b** Isotopic labeling. **c** Potential migratory process

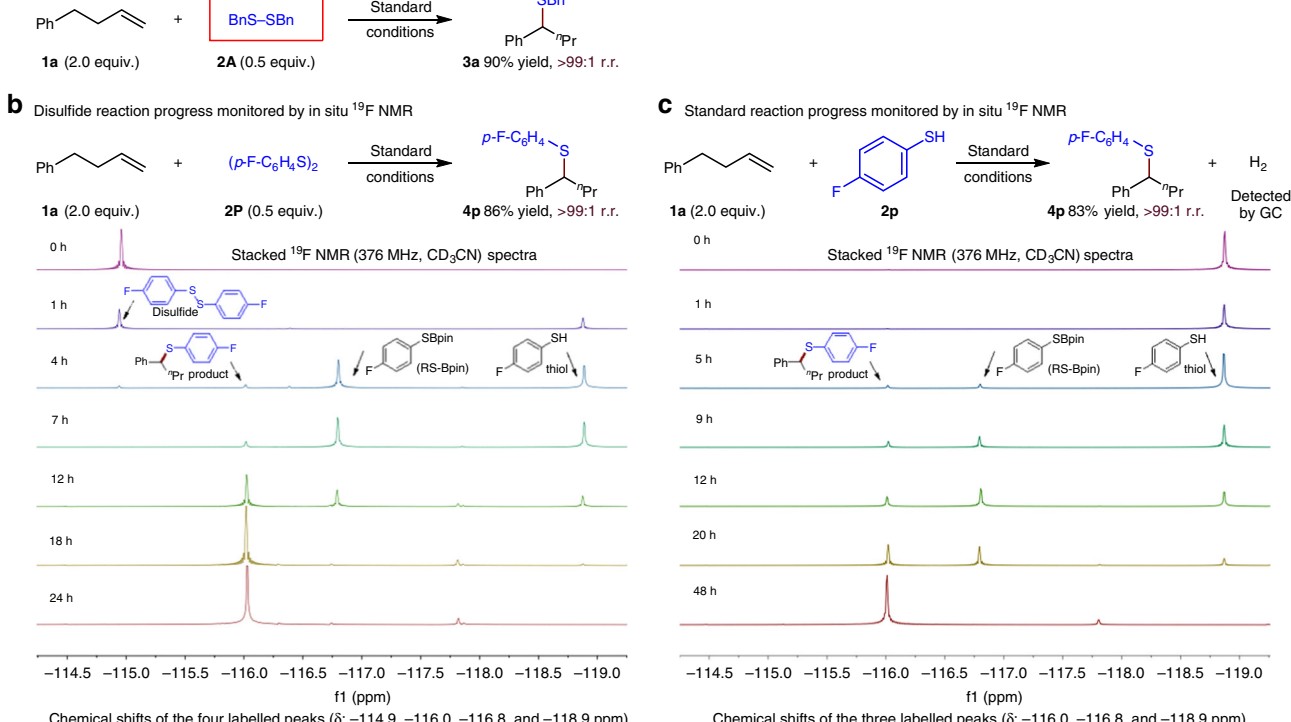

**Fig. 6** Potential reactive intermediate of thiol. **a** Potential intermediates: disulfide. **b** Disulfide reaction progress monitored by in situ fluorine-19 nuclear magnetic resonance ($^{19}$F NMR) (376 MHz, CD$_3$CN). **c** Standard reaction progress monitored by in situ $^{19}$F NMR (376 MHz, CD$_3$CN)

1:1 stoichiometry of alkene and RS-Bpin is used and the yields are even better (88 and 93%, respectively) when a slightly excess of alkene or RS-Bpin is used (1.2 equiv.). This finding, together with the results disclosed in Fig. 6, suggests that the reactive intermediate of thiol is the RS-Bpin complex.

Control experiments reveal that the solvent THF plays an important role. Only 5% yield of desired product is observed in the absence of this solvent. As shown in Fig. 8a, a different reactivity is observed when the solvent THF is replaced by a variety of other ethers in the standard reaction conditions. The nature of the ether backbone plays a crucial role, we found that only cyclic ethers with a β-hydride can produce the desired product in a reasonable yield. In contrast, acyclic ethers or cyclic ethers lacking a β-hydride do not have such a profound effect on reactivity. We postulated that the ether solvent might participate in the catalytic cycle. To verify this hypothesis, additional studies about the amount and consumption of ethers were carried out. As shown in Fig. 8b, both the yields and regioselectivities improved when the amount of THF is increased. The consumption of ether could also be observed during the reaction process (Fig. 8c). Only trace amounts of product (~1% yield) are produced during the first 6 h and the regioselectivity is poor in the first 12 h of the

reaction. Subsequently, the yield of desired product increases significantly, but the yield of other regioisomers (linear isomer) fails to increase after the first 12 h (Fig. 8c, entry 3 vs. entry 1). The origin of this apparent induction period as well as initial low regioselectivity is still under investigation. Finally, as shown in Fig. 4d, when deuterated THF-$d_8$ is used in both standard and modified standard reactions, a small amount of deuterium scrambling and deuterium incorporation at all positions except the benzylic position along the aliphatic chain of the desired product is observed by $^2$H NMR and mass spectrometric analysis. This indicates that a small amount of NiD is involved in the chainwalking process and the small amount of NiD should come from the deuterated THF-$d_8$.

To probe further the role of THF, Boron-11 NMR ($^{11}$B NMR) experiments were carried out to trace the standard reaction. As shown in Fig. 8e, the generation of RS-Bpin intermediate ($\delta = 33.6$ ppm) is confirmed again by $^{11}$B NMR spectroscopic analysis. We could also observe two new boron signals accompanied with the consumption of THF, which matches the signals of Bpin-nOBpin ($\delta = 21.2$ ppm) and ROBpin ($\delta = 22.3$ ppm).

Although an in-depth mechanistic discussion must await further investigation, a description of the proposed pathway, based on the above mechanistic studies, is shown in Fig. 9. The active nickel(I) hydride species (I)[61–66], which is initially formed from a Ni(II) precursor, a ligand, and an hydride source, inserts into the alkene (1a), and initiates the relatively fast and reversible chain-walking process through iterative β-hydride elimination/migratory reinsertion processes. A series of isomeric alkylnickel(I) species (II, IV, …) is then accessed through this chainwalking process. Controlled by the choice of ligand, selective reaction of the benzylic alkylnickel(I) intermediate (IV) with the thiolation reagent, the RS-Bpin (2A′) generated in situ from thiol and pinacolborane, probably through an oxidative addition and sequential facile reductive elimination[67–70] process then delivers the benzylic thiolation product (3a) along with LNi(I)Bpin (V). The active LNi(I)Bpin (V) species is then captured by THF to

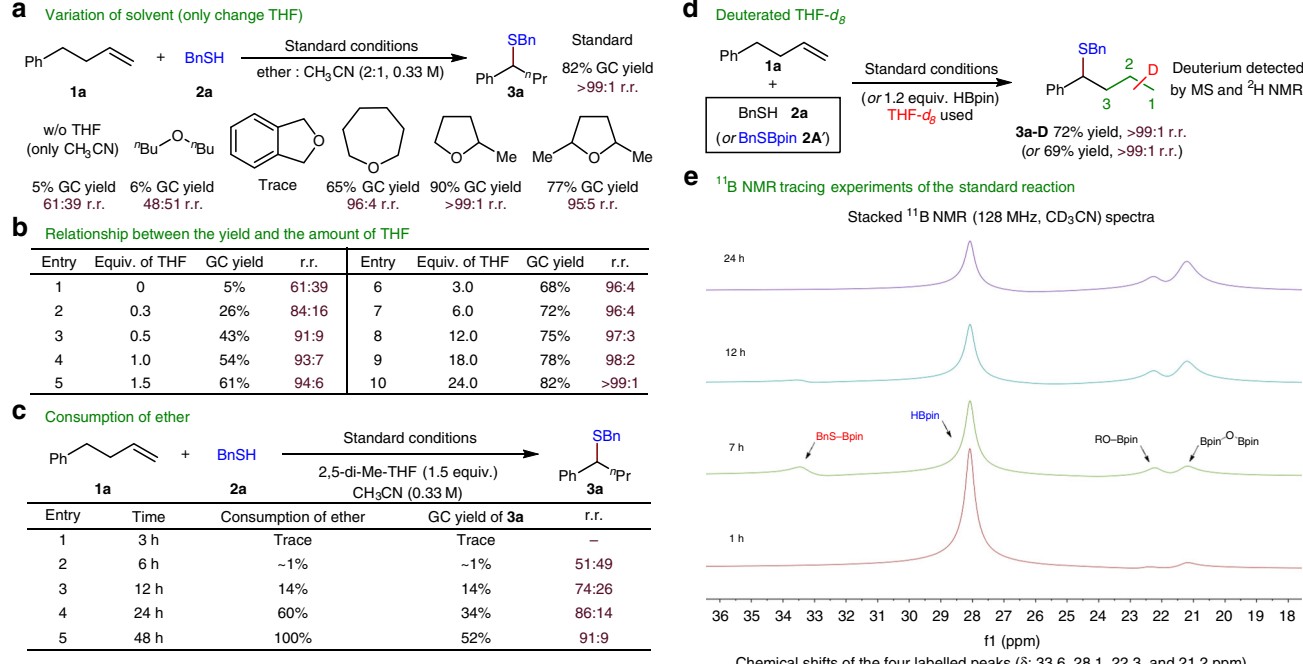

**Fig. 7** Optimized reaction conditions using RS–Bpin as the thiolation reagent. 0.20 mmol Scale, average of two experiments

| Entry | Deviation from above conditions | GC yield of 3a (%) | r.r. |
|---|---|---|---|
| 1 | None | 95 | >99:1 |
| 2 | (MeO)$_2$MeSiH, instead of HBpin | 40 | >99:1 |
| 3 | 1.0 equiv. HBpin | 95 | >99:1 |
| 4 | 1a : 2A′ = 1:1 | 84 | >99:1 |
| 5 | 1a : 2A′ = 1:1.2 | 88 | >99:1 |
| 6 | 1a : 2A′ = 1.2:1 | 93 | >99:1 |

**Fig. 8** Role of solvent tetrahydrofuran (THF). **a** Variation of solvent (only change THF). **b** Relationship between the yield and the amount of THF. **c** Consumption of ether. **d** Deuterated THF-$d_8$. **e** Reaction progress monitored by in situ Boron-11 NMR ($^{11}$B NMR) (128 MHz, CD$_3$CN) of the standard reaction

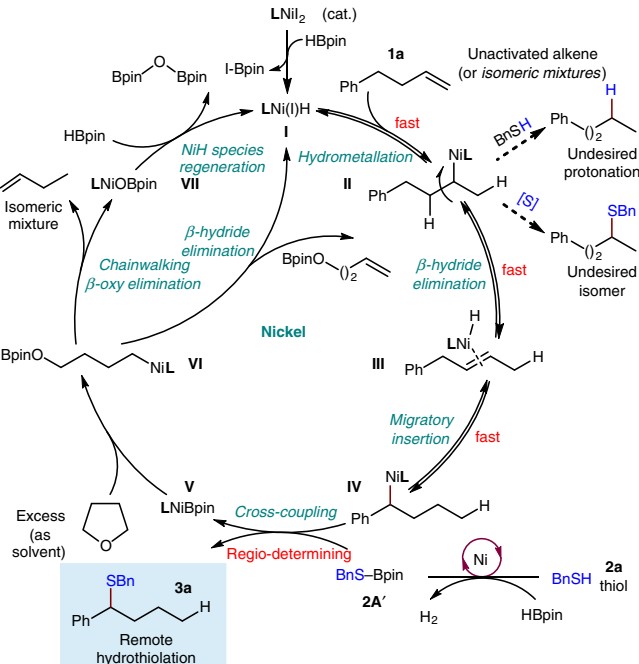

**Fig. 9** The catalytic cycle. Proposed mechanism for remote hydrothiolation. **L**, ligand

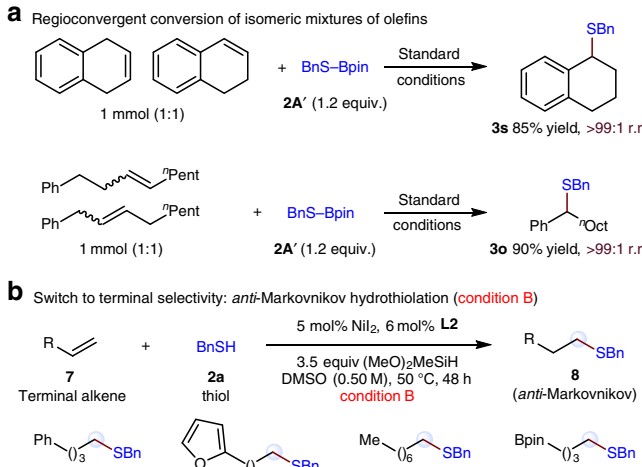

**Fig. 10** Regioconvergent and expanded site-selectivity experiments. **a** Regioconvergent conversion of isomeric mixtures of olefins. **b** Switch to terminal selectivity: *anti*-Markovnikov hydrothiolation

generate the corresponding alkylnickel(I) intermediate (**VI**). Sequential chainwalking and β-oxy elimination would deliver the isomeric mixture of butene along with LNi(I)OBpin (**VII**). The nickel hydride species (**I**) is then regenerated in situ by a stoichiometric amount of the pinacolborane to complete the catalytic cycle. Additional studies aimed at a full elucidation of the reaction pathway are in progress.

Mixtures of olefin isomers are generally more widely available than single isomers. Owing to the difficulty of isolation of each pure isomer, such mixtures are substantially cheaper than the pure isomers. Conversion of such mixtures in a regioconvergent process into value-added specialty chemicals is therefore of considerable interest. As expected, the robustness and utility of this catalytic system are further demonstrated through the employment as starting materials of isomeric mixtures of olefins, and the benzylic thioethers (**3s** and **3o**) can be obtained in high yield as a single regioisomer in both cases (Fig. 10a).

Finally, as shown in Fig. 2 and Fig. 10b, the current benzylic regioselectivity can also be switched to a terminal site to form the *anti*-Markovnikov hydrothiolation[54–60] products. A series of terminal alkenes can be effectively hydrothiolated under a modified reaction conditions (**8a–8d**).

In summary, we have developed a NiH-catalyzed remote hydrothiolation reaction of alkenes using thiols directly as thiolation reagents. This transformation utilizes readily accessible alkenes/alkynes and thiols as starting materials and earth-abundant nickel salts as catalysts. The mild process allows the direct installation of a thioether group at a benzylic, α-ether, or a terminal position with excellent regio- and chemoselectivity, as well as high functional group tolerance. Moreover, mechanistic studies reveal that the activated thiolation reagent is the RS-Bpin intermediate, and the ether solvent plays an important role in the regeneration of NiH species. Finally, the practical value of this transformation is highlighted by the regioconvergent conversion of unrefined isomeric mixtures of alkenes. The application of this protocol in cysteine bioconjugation as well as an asymmetric version of the current transformation is currently in progress and will be reported in due course.

## Methods

**General procedure for NiH-catalyzed remote hydrothiolation**. To an oven-dried 8 mL screw-cap vial equipped with a magnetic stir bar was added NiI₂ (3.2 mg, 5.0 mol%) and bathocuproine (**L1**, 2,9-dimethyl-4,7-diphenyl-1,10-phenanthroline) (4.0 mg, 6.0 mol%). The vial was introduced into a nitrogen-filled glove box, anhydrous THF (0.40 mL) and CH₃CN (0.20 mL) were added, and the mixture was stirred for 10 min, at which time alkene (0.40 mmol, 2.0 equiv.), benzyl mercaptan (25.0 mg, 0.20 mmol, 1.0 equiv.), HBpin (pinacolborane, 100 μL, 0.70 mmol, 3.5 equiv.) and Li₃PO₄ (50 mg, 0.40 mmol, 2.0 equiv.) were added to the resulting mixture in this order. The tube was sealed with a teflon-lined screw cap, removed from the glove box and stirred at 60 °C for 24 h (the mixture was stirred at 750 rpm). After the reaction was complete, the reaction mixture was immediately filtered through a short pad of silica gel (using EtOAc in hexanes) to give the crude product. *n*-Tetradecane (20 μL) was added as an internal standard for GC analysis. 1,1,2,2-Tetrachloroethane (10.5 μL, 0.10 mmol) was added as internal standard for ¹H NMR analysis of the crude material. The product was purified by chromatography on silica gel for each substrate. The yields reported are the average of at least two experiments, unless otherwise indicated. See Supplementary Information for more detailed experimental procedures and characterization data for all products.

## Data availability

The authors declare that the main data supporting the findings of this study, including experimental procedures and compound characterization, are available within the article and its Supplementary Information files, or from the corresponding author upon reasonable request.

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

## Acknowledgements

Research reported in this publication was supported by NSFC (21822105, 21772087, and 21602101), NSF of Jiangsu Province (BK20160642). We are grateful to Prof. Yan Zhu (NJU) for GC analysis of H$_2$ and to Prof. Xiaoliang Yang (NJU) for NMR analysis.

## Author contributions

Y.Z. and S.Z. designed the project. Y.Z., X.X., and S.Z. co-wrote the manuscript, analyzed the data, discussed the results, and commented on the manuscript. Y.Z. and X.X. performed the experiments. All authors contributed to discussions.

## Additional information

**Competing interests:** The authors declare no competing interests.

