## [Peer Review File · Nature Communications]

Reviewers' comments:

Reviewer #1 (Remarks to the Author):

The manuscript by Zhu and co-workers describes an interesting process of a NiH-catalysed remote hydrothiolation of alkenes/alkynes with thiols, which is sure to attract significant attention in the organic synthesis and catalysis communities. Compared with previous work, this process avoids the preparation of electrophilic coupling reagents and could selectively couple with thiol group in the presence of other nucleophiles. With the solvent THF playing into the game, the mechanistic investigation is well-conducted and very impressive. As well as being mechanistically stimulating, the method appears practical and robust, and I point to products 4f-4h, 4y, 4z, and 4a' as particularly interesting results. Overall, the reported chemistry represents a novel and synthetically valuable merger of metal hydride-mediated chainwalking and selective C-S coupling, and would deserve publication as an article in Nature Communications. The authors are suggested to address the following points in a revised manuscript.

1. In the Table 2, is 1,1-disubstituted terminal alkene also a suitable substrate?
2. Could thiocarboxylic acid be a suitable coupling partner?
3. Although it is not necessary at current stage, it is interesting to know if it is possible to run the current reaction in an enantioselective way.
4. In the Table 2c, the fluorine in the "4-fluorostyrene" should be corrected to "R"; in the line 136, "Tabel 3" is misspelt.
5. The following article should be cited in the context of the NiH-catalyzed remote functionalisation: Communications Chemistry 2019, 2, 5.
6. In SI, a caption should be included on the NMR spectrum, noting the nucleus being measured, the solvent (formula preferred, e.g. C₆D₆ over benzene-d₆), and the field strength. Please amend.

Reviewer #2 (Remarks to the Author):

This manuscript titled, Nickel-catalysed selective migratory conjugation of alkenes/alkynes with aliphatic and aromatic thiols by Shaolin Zhu and co-workers is one of the best if not the best thiolation methodology developed in the past 20 years! The method is simple, not light dependent, practical, useful, general, and has a great potential to impact drugs discovery and medicine design in pharmaceutical research. The mechanistic studies and subsequent proposed mechanism are also spot on. This work provides many relevant insights and it will certainly promote the development of new innovative and novel methods. In addition to the exceptional broad scope of this method and its remarkable functional groups tolerance as well as its high regioselectivity, the supporting information is also very detailed. Put together, this is a very useful and highly important novel methodology. This manuscript is suitable for publication in Nature Communications and I strongly recommend its acceptance as is.

Reviewer #3 (Remarks to the Author):

Zhu and co-workers report on Ni-catalyzed migratory hydrothiolation of alkenes and alkynes. It is a novel strategy for the selective C-S bond formation in the benzylic position with a broad substrate scope and preliminary mechanistic studies. While I am not much convinced about the practicality of this methodology (alkene is used in excess; need of overstoichiometric additives; by-product, which needs to be removed), it still possesses a high level of novelty and it might open doors to new development of important sulfur-containing compounds. Therefore, I suggest to accept this work, however after addressing following questions and remarks:

1. Title: I wouldn't call this reaction "migratory conjugation"
2. Graphical abstract: please simplify, there is too much information. I don't understand the sign with crossed out "E". To me, it is obvious that no electrophilic reagents are needed for hydrothiolation reaction. I would also change the text correspondingly.
3. In the introduction, the authors state that the thiolation is non-directed. I don't agree, since the reaction occurs at the benzylic position, which is not non-activated.
4. The authors state that the transformation is regioconvergent. They show several times that an anti-Markovnikov functionalization can occur at the terminal position of the olefin (table 1, figure 6), it is also mentioned many times. Did the authors perform control experiments in the absence of catalyst? This is a well known reaction, sometimes referred to as thiol-ene reaction, which does not require any catalyst and can be promoted by traces of oxygen or electron transfer: (Ber. Dtsch. Chem. Ges. 1905, 38, 646; <https://doi.org/10.1002/ejoc.201701692>; <https://doi.org/10.1002/chem.201103252>). The authors should perform the control experiments and remove the speculations.
5. Page 1, line 44: References 16-49: isomerizing olefin transformations are not that new, especially hydroformylation or alkoxy-carbonylation are well established processes and should be cited.
6. Table 2: The scope is quite broad, but no ortho-substituted aryl moieties were used (the authors should try at least 2 more such substrates). What about tetrasubstituted olefins?

7. Table 3: Why are the reaction conditions different (more catalyst, different solvent, longer reaction time)
8. What is the role of phosphate?
9. Are all reaction components dissolved in the reaction mixture?
10. Page 9, line 170: Why is there no D in the benzylic position?
- 11- Figure 2: How is compound 5 formed?
12. Page 12, line 212: catalytic amount of product does not exist. Change to 5%.
13. Figure 4c: why is the yield lower and regioselectivity worse here?
14. Page 12: Are there any products from THF observed?
15. Figure 5 describes two pathways for the final steps of the catalytic cycle. Was butene or the beta-hydride elimination product observed?
16. Page 15, lines 269-270: It is not true that anti-Markovnikov hydrothiolation is a challenge.
17. Figure 6: Were the alkene mixtures synthesized or were they mixed together?
18. Page 16: Why are 2 equivalents of alkene required? This is a big disadvantage of the method! Did the authors ever encounter separation problems?
19. The reaction should be demonstrated on larger scope, at least for two examples.

In response to **reviewer 1** (quotes from reviewer are italicized):

Reviewer #1 (Remarks to the Author):

The manuscript by Zhu and co-workers describes an interesting process of a NiH-catalysed remote hydrothiolation of alkenes/alkynes with thiols, which is sure to attract significant attention in the organic synthesis and catalysis communities. Compared with previous work, this process avoids the preparation of electrophilic coupling reagents and could selectively couple with thiol group in the presence of other nucleophiles. With the solvent THF playing into the game, the mechanistic investigation is well-conducted and very impressive. As well as being mechanistically stimulating, the method appears practical and robust, and I point to products 4f-4h, 4y, 4z, and 4a' as particularly interesting results. Overall, the reported chemistry represents a novel and synthetically valuable merger of metal hydride-mediated chainwalking and selective C-S coupling, and would deserve publication as an article in Nature Communications. The authors are suggested to address the following points in a revised manuscript.

1. In the Table 2, is 1,1-disubstituted terminal alkene also a suitable substrate?

1,1-Disubstituted terminal alkene is not a suitable substrate for the remote hydrothiolation in our current condition. Both the yield and r.r. are not good. We have added this result as one of the unsuccessful substrates in the supplementary information (see page S46, Supplementary Scheme 2).

2. Could thiocarboxylic acid be a suitable coupling partner?

Thiocarboxylic acid is not a suitable substrate in our current condition. We could not observe any coupling product. We have added this result as one of the unsuccessful substrates in the supplementary information (see page S46, Supplementary Scheme 2).

3. Although it is not necessary at current stage, it is interesting to know if it is possible to run the current reaction in an enantioselective way.

Currently, using a chiral pyrox ligand has demonstrated a not excellent level of enantioselectivity (13% ee, see page S45, Supplementary Scheme 1). Progress in this area will be reported in due course. We have mentioned this information in the main text (at the end of conclusion section).

4. In the Table 2c, the fluorine in the “4-fluorostyrene” should be corrected to “R”; in the line 136, “Tabel 3” is misspelt.

We have fixed several typos.

5. The following article should be cited in the context of the NiH-catalyzed remote functionalisation: Communications Chemistry 2019, 2, 5.

We have cited the above-mentioned paper (see: ref. 52).

6. In SI, a caption should be included on the NMR spectrum, noting the nucleus being measured, the solvent (formula preferred, e.g. C6D6 over benzene-d6), and the field strength. Please amend.

We have now added the above-mentioned information in our NMR spectrum.

In response to **reviewer 2** (quotes from reviewer are italicized):

Reviewer #2 (Remarks to the Author):

This manuscript titled, Nickel-catalysed selective migratory conjugation of alkenes/alkynes with aliphatic and aromatic thiols by Shaolin Zhu and co-workers is one of the best if not the best thiolation methodology developed in the past 20 years! The method is simple, not light dependent, practical, useful, general, and has a great potential to impact drugs discovery and medicine design in pharmaceutical research. The mechanistic studies and subsequent proposed mechanism are also spot on. This work provides many relevant insights and it will certainly promote the development of new innovative and novel methods. In addition to the exceptional broad scope of this method and its remarkable functional groups tolerance as well as its high regioselectivity, the supporting information is also very detailed. Put together, this is a very useful and highly important novel methodology. This manuscript is suitable for publication in Nature Communications and I strongly recommend its acceptance as is.

We thank reviewer 2 for the positive comments.

In response to **reviewer 3** (quotes from reviewer are italicized):

Reviewer #3 (Remarks to the Author):

Zhu and co-workers report on Ni-catalyzed migratory hydrothiolation of alkenes and alkynes. It is a novel strategy for the selective C-S bond formation in the benzylic position with a broad substrate scope and preliminary mechanistic studies. While I am

not much convinced about the practicality of this methodology (alkene is used in excess; need of overstoichiometric additives; by-product, which needs to be removed), it still possesses a high level of novelty and it might open doors to new development of important sulfur-containing compounds. Therefore, I suggest to accept this work, however after addressing following questions and remarks:

1. Title: I wouldn't call this reaction "migratory conjugation"

Changed.

2. Graphical abstract: please simplify, there is too much information. I don't understand the sign with crossed out "E". To me, it is obvious that no electrophilic reagents are needed for hydrothiolation reaction. I would also change the text correspondingly.

Changed.

3. In the introduction, the authors state that the thiolation is non-directed. I don't agree, since the reaction occurs at the benzylic position, which is not non-activated.

Changed.

4. The authors state that the transformation is regioconvergent. They show several times that an anti-Markovnikov functionalization can occur at the terminal position of the olefin (table 1, figure 6), it is also mentioned many times. Did the authors perform control experiments in the absence of catalyst? This is a well known reaction, sometimes referred to as thiol-ene reaction, which does not require any catalyst and can be promoted by traces of oxygen or electron transfer: (Ber. Dtsch. Chem. Ges. 1905, 38, 646; <https://doi.org/10.1002/ejoc.201701692>; <https://doi.org/10.1002/chem.201103252>). The authors should perform the control experiments and remove the speculations.

We have performed the control experiments of the *anti*-Markovnikov thiolation in the absence of catalyst and do observe a small amount of background reaction (25% yield). The yield of background reaction is not as good as that in our reaction conditions. We have now added this information in the supplementary information (see page S75, Supplementary Table 8 and 9) and also change the main text correspondingly. We also have cited the reference mentioned (see: ref. 69).

5. Page 1, line 44: References 16-49: isomerizing olefin transformations are not that new, especially hydroformylation or alkoxyacylation are well established processes and should be cited.

We have cited the reference mentioned (see: refs. 19-21).

6. Table 2: The scope is quite broad, but no *ortho*-substituted aryl moieties were used (the authors should try at least 2 more such substrates). What about tetrasubstituted olefins?

We have now added two more examples with *ortho*-substituted aryl moieties (**3g** and **3h**).

The isomerization is blocked when tetrasubstituted olefins were used. No desired alkenes isomerization-hydrothiolation product was observed. We have added this result as one of the

unsuccessful substrates in the supplementary information (see page S46, Supplementary Scheme 2).

7. Table 3: Why are the reaction conditions different (more catalyst, different solvent, longer reaction time)

Benzylic thiol is more reactive relative to other thiols. While other thiols were used, we modified the reaction condition in order to get better yields (Table 3).

8. What is the role of phosphate?

Although phosphate is not essential for the reaction to proceed, it is beneficial for the yield. Control experiment reveals that the amount of corresponding byproduct alkane is reduced when phosphate is used. Currently, the exact in-depth role of phosphate is still under investigation.

9. Are all reaction components dissolved in the reaction mixture?

All the reaction components are dissolved in the reaction mixture except the inorganic base Li_3PO_4 .

10. Page 9, line 170: Why is there no D in the benzylic position?

Under the standard reaction conditions, the NiD prefer to insert into styrene in a highly regioselective way to generate only benzylic alkylnickel species.

11- Figure 2: How is compound 5 formed?

We propose that the compound **5** is formed through the homocoupling of two benzylic radicals. A comment has been added in Supplementary Information Page 47.

12. Page 12, line 212: catalytic amount of product does not exist. Change to 5%.

Changed.

13. Figure 4c: why is the yield lower and regioselectivity worse here?

Here we used only 1.5 equiv. 2,5-di-Me-THF instead of 0.40 mL THF in Figure 4c. The use of excess amount of ether is crucial for the obtaining product in high yield and r.r., probably because part of ether is consumed by NiH directly and then 1.5 equiv. ether is insufficient for the desired conversion. In contrast (Figure 4a), when 0.40 mL 2,5-di-Me-THF is used, the yield and r.r. is better.

14. Page 12: Are there any products from THF observed?

In accord with this comment, as well as the comment 15, we have now added the following information in the supplementary information. Due to the low boiling point of butene, we are not able to monitor it by the ordinary GC and GC/MS in our lab. Studies in this regard are still under investigation in collaboration with other lab. We hope to address this issue in the future. However, we do observe the beta-hydride elimination product by GC and GC/MS in the form of alcohol (see page S69, Supplementary Scheme 18 and Supplementary Figure 17).

15. Figure 5 describes two pathways for the final steps of the catalytic cycle. Was butene or the beta-hydride elimination product observed?

In accord with this comment, as well as the comment 14, we have now added the following information in the supplementary information. Due to the low boiling point of butene, we are not able to monitor it by the ordinary GC and GC/MS in our lab. Studies in this regard are still under investigation in collaboration with other lab. We hope to address this issue in the future. However, we do observe the beta-hydride elimination product by GC and GC/MS in the form of alcohol (see page S69, Supplementary Scheme 18 and Supplementary Figure 17).

16. Page 15, lines 269-270: In is not true that anti-Markovnikov hydrothiolation is a challenge.

Changed.

17. Figure 6: Were the alkene mixtures synthesized or were they mixed together?

As a proof of concept, we only mixed the alkenes to obtain a mixture.

18. Page 16: Why are 2 equivalents of alkene required? This is a big disadvantage of the method! Did the authors never encounter separation problems?

GC and GC/MS analysis indicates that part of alkene is reduced to the byproduct alkane under the standard reaction conditions. As show in our proposed reaction pathway (Figure 5), this byproduct is formed through the protonation of alkylnickel species (the proton source is RSH). When RS-Bpin (there is no proton source) is used instead of thiol, only a 1:1 ratio of alkene and thiolation reagent (RS-Bpin) is needed to obtain good yield.

The separation of the byproduct alkane and our desired thioether is not a difficult problem.

19. The reaction should be demonstrated on larger scope, at least for two examples.

We have now added two extra substrates (**3j** and **3k** in Table 2), which are derived from complex molecules.

Thank you for editing our manuscript. I hope that you find the revised manuscript suitable for publication in the *Nature Communications*. Please do not hesitate to contact me if you have further comments to discuss on any of these points.

REVIEWERS' COMMENTS:

Reviewer #1 (Remarks to the Author):

I am OK with the revision.

Reviewer #3 (Remarks to the Author):

The revision is very thorough and detailed. The authors answered all of my questions and also those of other reviewers. I hope to be able to follow up on this chemistry in the future. The manuscript should be accepted in this form.